# Impact of Combined Prebiotic Galacto-Oligosaccharides and *Bifidobacterium breve*-Derived Postbiotic on Gut Microbiota and HbA1c in Prediabetic Adults: A Double-Blind, Randomized, Placebo-Controlled Study

**DOI:** 10.3390/nu16142205

**Published:** 2024-07-10

**Authors:** Beyda Beteri, Monica Barone, Silvia Turroni, Patrizia Brigidi, George Tzortzis, Jelena Vulevic, Karol Sekulic, Diana-Elena Motei, Adele Costabile

**Affiliations:** 1School of Life and Health Sciences, University of Roehampton, London SW15 4JD, UK; beyda.beteri@roehampton.ac.uk (B.B.); diana-elena.motei@roehampton.ac.uk (D.-E.M.); 2Human Microbiomics Unit, Department of Medical and Surgical Sciences, University of Bologna, 40138 Bologna, Italy; monica.barone@unibo.it (M.B.); p.brigidi@unibo.it (P.B.); 3Unit of Microbiome Science and Biotechnology, Department of Pharmacy and Biotechnology, University of Bologna, 40126 Bologna, Italy; silvia.turroni@unibo.it; 4veMico Ltd., Amelia House, Crescent Road, Worthing BN11 1RL, UK; george@vemico.co.uk (G.T.); jelena@vemico.co.uk (J.V.); 5Alberta Health Services, Edmonton, AB T5J 3E4, Canada; karolina.sekulic@albertahealthservices.ca

**Keywords:** *Bifidobacterium breve*, prediabetes, gut microbiome, postbiotics

## Abstract

The complex interactions between intestinal microbiota and metabolic disorders are well-documented, with implications for glucose metabolism, energy expenditure, and intestinal permeability. Prebiotics induce beneficial changes in gut microbiota composition in prediabetes, while postbiotics can enhance gut barrier function, complementing each other to improve glucose metabolism and insulin sensitivity. This study investigated the effects of a 12-week dietary fibre (DF) supplement on gut health, metabolic function, and diet. The supplement contained konjac glucomannan (KGM), galacto-oligosaccharides (GOSs), and exopolysaccharides (EPSs) from *Bifidobacterium breve*. In a randomised, double-blind, placebo-controlled, parallel-group clinical trial, 53 prediabetic volunteers were randomly assigned to either a daily DF supplement (YMETA) or a placebo (cellulose microcrystalline) for 12 weeks, followed by a 4-week follow-up. Measurements included gut microbiota composition, glycated haemoglobin (HbA1c), fasting plasma glucose (FPG), plasma lipids, anthropometry, body composition, blood pressure, and dietary intake. The intervention group showed a significant increase in alpha diversity and butyrate-producing bacteria, with reductions in HbA1c and FPG levels below prediabetes thresholds. No significant changes were observed in the placebo group. This study suggests that manipulating the human gut microbiome through dietary interventions could be a promising therapeutic approach to managing prediabetes and preventing or delaying diabetes.

## 1. Introduction

Prediabetes stands as a critical health concern, affecting an estimated 5.8% (298 million) of adults aged 20–79 with impaired fasting glucose (IFG) and 9.1% (464 million) with impaired glucose tolerance (IGT) worldwide [1]. This metabolic condition, characterised by elevated blood glucose levels, reflects underlying insulin resistance or inadequate pancreatic insulin secretion [2]. As an intermediate stage between normoglycemia and type 2 diabetes mellitus (T2DM) [3], prediabetes poses a significant risk factor for progressing to overt T2DM if left unmanaged, contributing to the global burden of diabetes-related complications [2]. Despite its clinical importance, diagnosing prediabetes remains challenging due to its often-asymptomatic nature, leading to underdiagnosis and delayed intervention [2]. Moreover, comprehensive epidemiological data on prediabetes are lacking globally, underscoring the need for enhanced surveillance to inform preventative strategies [1,2].

The escalating prevalence of prediabetes mirrors the concurrent rise in diabetes incidence, necessitating effective interventions to curb disease progression [1]. Diabetes was among the top 10 causes of death globally in 2019, claiming 1.5 million lives annually, with the majority attributed to T2DM [4]. While conventional pharmacotherapy, notably metformin and insulin, forms the cornerstone of diabetes management, concerns persist regarding its long-term efficacy and potential adverse effects, including perturbations in gut microbiota composition, their metabolic activity, and inflammation [5,6]. Growing evidence implicates gut microbiota dysbiosis in the pathogenesis of metabolic disorders, including obesity and diabetes, highlighting the intricate interplay between the gut microbiota, host metabolism, and immune function [6,7]. Dysbiosis disrupts metabolic homeostasis by promoting inflammation, impairing the intestinal barrier, altering bile acid metabolism, and affecting the production of metabolites like short-chain fatty acids (SCFAs) [6,7]. These alterations exacerbate insulin resistance and glucose intolerance, contributing to the pathogenesis of prediabetes and T2DM [7]. In this context, dietary interventions targeting gut microbiota represent a promising avenue for prediabetes management and T2DM prevention. Prebiotic and soluble fibre supplements offer a non-pharmacological approach to modulate gut microbiota composition and activity and improve metabolic health. These supplements promote the growth of beneficial bacteria, but may also mitigate inflammation, enhance barrier function, and ameliorate glucose intolerance [8,9]. This study aimed to investigate the therapeutic potential of a 12-week dietary-fibre (DF) supplement, YMETA, comprising galacto-oligosaccharides (GOSs) produced through the β-galactosidase activity of *Bifidobacterium breve*, exopolysaccharides (EPSs) from *B. breve*, and konjac glucomannan (KGM) in prediabetic individuals aged 18–60 years. Firstly, we conducted a comprehensive assessment of gut health, including gut microbiota composition and intestinal permeability, as well as metabolic function, specifically glycated haemoglobin (HbA1c) and fasting plasma glucose (FPG) levels. Then, we evaluated alterations in anthropometry, body composition, blood pressure (BP), and plasma lipids [total cholesterol (TC), HDL-C, LDL-C, triacylglycerol (TAG), non-HDL-C, and TC to HDL-C ratio (THR)]. Additionally, we examined total protein and diet to elucidate the impact of DF supplementation on prediabetes management and to inform evidence-based preventative strategies.

## 2. Materials and Methods

### 2.1. Study Design and Ethics

The present randomised, double-blind, placebo-controlled, parallel-group clinical trial aimed to investigate the effect of a DF supplement called YMETA (veMico Ltd., West Sussex, UK) on HbA1c and FPG levels in individuals with prediabetes. The study was conducted at the Health Science Research Centre, Department of Life Sciences, University of Roehampton, London, UK between June 2022 and August 2023, in accordance with the principles of the Declaration of Helsinki (2013). This trial is registered with the US National Library of Medicine Clinical Trials Registry (Identifier: NCT05400525; Study ID: LSC 22/374). The study protocol was approved by the Institutional Review Board of the University of Roehampton (Ethics Approval Ref: LSC 22/374), and written informed consent was obtained from all participants enrolled.

### 2.2. Inclusion and Exclusion Criteria

Inclusion criteria were adults aged 18 to 60 years with a fasting plasma glucose (FBG) level of 5.6 to 6.9 mmol/L or glycated haemoglobin (HbA1c) level of 5.7 to 6.4% (based on the ADA criteria), as determined by a baseline screening test. Participants were considered prediabetic if at least one of their blood samples met these criteria. Eligible participants were required to have a mobile phone and be proficient in English. General health was assessed using a pre-study medical questionnaire, and individuals were excluded if they met any of the following criteria: (i) current diagnosis or clinical history of T2DM or endocrine disorders; (ii) comorbid conditions such as a history of an acute cardiovascular event, uncontrolled hypertension, cancer, or major psychiatric or cognitive problems; (iii) participation in a weight loss programme; (iv) receiving medical treatment for prediabetes that affects glucose metabolism long term (e.g., corticosteroids); (v) elevated liver enzymes (alanine aminotransferase ≥ 300 IU/L, aspartate aminotransferase ≥ 300 IU/L); (vi) use of antibiotics or bacterial agents (probiotics) within 1 month; (vii) history of alcohol or drug abuse; or (viii) pregnant or breastfeeding women, women planning to become pregnant in the next 6 months, or women of childbearing potential not using contraception.

### 2.3. Study Participants

Two hundred and four volunteers were recruited from the local community and social media platforms such as Facebook or Instagram, which were widely used among adults. After completing a medical questionnaire at the first visit (baseline) to assess study eligibility, 53 volunteers (intervention: *n* = 25; placebo: *n* = 28) met the inclusion criteria and agreed to participate, while 151 volunteers were excluded. Five volunteers in the intervention group and three in the placebo group dropped out due to GI symptoms and personal issues but were screened for baseline.

### 2.4. Intervention

Fifty-three prediabetic volunteers (intervention: *n* = 25; placebo: *n* = 28) underwent three study visits at the University of Roehampton: baseline (visit 1), week 12 (visit 2), and week 16 (visit 3, washout period and study completion). Eligible participants were randomised to the intervention or placebo groups (Figure 1). Intervention group participants took a DF supplement, YMETA (veMico Ltd., UK), containing GOSs (60%) synthesised with β-glactosidase from *Bifidobacterium breve* 091109, EPSs (8%) from *B. breve* 091109, and KGM (32%). Cellulose microcrystalline (Alfa Aesar, Heysham, Lancashire, UK), a common excipient in the pharmaceutical industry known for its lack of impact in the colonic environment, was chosen as the placebo. Both supplements were provided in sachets containing 5 g for daily consumption over 12 weeks. Participants were advised to avoid consuming the supplement with hot beverages or food to maintain product consistency. They could consume the supplement at any time of the day, preferably in the evening if discomfort (e.g., bloating) occurred after morning intake. During the 12-week intervention, participants were contacted to monitor progress, ensure compliance, assess adverse symptoms, and schedule visits. They were reminded to fast for 10 h, refrain from smoking or chewing gum for 10 h, avoid exercising for 12 h, abstain from alcohol for 24 h, and increase their water intake before visits.

### 2.5. Outcomes

#### 2.5.1. Samples Collection for Blood Lipids, Immune/Inflammatory Markers, and Gut Microbiota Assessments

A 40 mL blood sample was collected by a phlebotomist at baseline, week 12, and week 16 visits for biochemical analysis. Blood samples were collected in plain, Lithium Heparin, EDTA, and Sodium Fluoride/Potassium Oxalate collection tubes (BD Vacutainer^®^ Cowley, Oxon., UK). All samples were kept on ice until centrifugation. Plasma samples were recovered by centrifugation at 1700 revolutions per minute for 10 min at 4 °C, dispensed into 1.5 mL microcentrifuge tubes, and frozen at −80 °C. Clinical chemical variables (i.e., HDL-cholesterol, HDL-C; LDL-cholesterol, LDL-C; total cholesterol, TAG; and the ratio of HDL-C to total cholesterol (TC)) were measured using Daytona Bioanalyser (Randox Laboratories Ltd., Antrim, UK). Faecal samples were collected using OMNIgene GUT self-collection tubes (DNA Genotek, Ottawa, Canada) at baseline (week 0), at the end of the intervention period (week 12), and after the washout period (week 16), and stored at −80 °C until further analysis.

#### 2.5.2. Glucose Biomarkers

Glucose biomarkers, HbA1c and FBG, were tested from whole blood samples collected in plain blood tubes. HbA1c was measured using NycoCard™ READER II (Abbott Laboratories, Copenhagen, Denmark) and FBG was measured using the Biosen C-Line Clinic (EKKF-diagnostic GmbH, Barleben, Germany). The Homeostatic Model Assessment for Insulin Resistance (HOMA-IR) value was calculated using the formulae [HOMA-IR = fasting insulin (mU/L) × glucose (nmol/L)/22.5]. Plasma insulin concentrations were measured using a commercially available human insulin assay (V-Plex Human Insulin Kit, Meso Scale Discovery (MSD) Rockville, MD, USA) as per manufacturer recommendations.

#### 2.5.3. Intestinal Permeability

Plasma Lipopolysaccharide (LPS) concentration was measured by Enzyme-Linked ImmunoSorbent Assay (ELISA) using Quantikine^®^ ELISA kits (R&D Systems Inc., Barton, UK) as detailed in Costabile et al. [10].

#### 2.5.4. Gut Microbiota Profiling

Microbial DNA was extracted from 250 mg of faecal samples using the QIAamp DNA Stool Kit (QIAGEN, Hilden, Germany) according to the manufacturer’s instructions. The DNA concentration was measured on a NanoDrop-One spectrophotometer (NanoDrop Technologies, Thermo Scientific, Waltham, MA, USA). The 16S rRNA gene sequencing and bioinformatics were conducted as previously reported [11]. In summary, PCR amplification of the hypervariable V3-V4 regions of the 16S rRNA gene used the S-D-Bact-0341-b-S-17/S-D-Bact-0785-a-A-21 primers [12] incorporating Illumina overhang adapter sequences as per the manufacturer’s instructions. PCR products were subsequently purified using magnetic-bead-based technology (Agencourt AMPure XP; Beckman Coulter, Brea, CA, USA), followed by indexing via limited-cycle PCR utilizing Nextera technology and subsequent purification. Final libraries were then pooled at equimolar concentrations (4 nM), denatured using 0.2 N NaOH, and diluted to 6 pM prior to loading onto the MiSeq flow cell. Sequencing was conducted on an Illumina MiSeq platform using a 2 × 250 bp paired-end protocol according to the manufacturer’s specifications (Illumina, San Diego, CA, USA). The sequencing reads have been deposited in the National Center for Biotechnology Information Sequence Read Archive (Bioproject ID PRJNA1131172).

#### 2.5.5. Anthropometric and Body Composition Analyses

Anthropometric and body composition variables were monitored at each visit. Participants were asked to remove shoes, socks, heavy clothing, and heavy items. Height was measured using a portable stadiometer (Leicester Height Measure; Seca Ltd., Birmingham, UK) with participants standing with their feet together, arms at their sides, shoulders straight, heels, buttocks, and upper back touching the stadiometer, and head in the Frankfort position. Waist and hip circumference were measured using an anthropometric tape (Seca 201, Hamburg, Germany) over a thin layer of clothing with the subjects in a relaxed position with their feet together. Waist circumference was measured at the midpoint between the lowest rib and the anterior iliac crest. Hip circumference was measured at the largest posterior protuberance of the buttocks. WHR was calculated as waist circumference divided by hip circumference. All anthropometric measurements were rounded to one decimal place.

Body composition (weight, Body Mass Index (BMI), body fat% and fat mass) was assessed using the Tanita BC-418 MA Segmental Body Composition Analyser (Tanita Corporation, Tokyo, Japan). The device was calibrated to account for clothing weight (0.5 kg). Weight was recorded after entering the subject’s age, height, gender, and activity level (“standard” or “athlete”). All participants of this study were classified as standard.

#### 2.5.6. Blood Pressure (BP)

Systolic and diastolic blood pressure (BP) (mmHg) were measured at each study visit using a digital BP monitor (Nissei, model DS-1902, Japan Precision Instruments, Inc., Gunma, Japan) with participants comfortably seated and the arm supported at heart level.

#### 2.5.7. Dietary Intake and Analyses

Participants recorded their dietary intake for 4 days (3 weekdays and 1 weekend day) at the start of the intervention. Dietary analyses were performed by a nutritionist using Nutritics (Research Edition, v5.096, Dublin, Nutritics, 2019). For missing details on food type or portion, default types and portions from the Nutritics database were used. For food brands not included in the database, equivalent foods were selected.

#### 2.5.8. Gastrointestinal Changes (GI)

Participants recorded GI habits (bowel movements, stool patterns, abdominal pain, bloating, and gas) and mood changes (happiness, alertness, energy, and stress) throughout the treatment and washout periods. Bowel movements were recorded as the number of stools per day and stool consistency using the Bristol stool chart (score 1 to 7). Abdominal pain, bloating and flatulence were ranked from 0 (no symptoms), 1 (mild), 2 (moderate), to 3 (severe symptoms). Mood changes were recorded on a scale of 0 (less than normal), 1 (normal), to 2 (more than normal).

#### 2.5.9. Statistical Analysis

Data were assessed for outliers, missing data, and normality using the Kolmogorov–Smirnov test. Data were expressed as mean ± SD. Baseline characteristics were analysed using the Independent (Unpaired) Samples *t*-Test. Within-group biochemical analysis and body composition results were analysed using the Dependent (Paired) *t*-Test (i.e., before vs. after intervention) and Independent Samples *t*-Test was used for group comparison analysis (intervention versus placebo). Statistical analyses were performed using IBM SPSS Statistics (Computer software, version 28, SPSS Inc., Chicago, IL, USA) [13].

For microbiota data, statistical analyses were conducted using RStudio 1.0.44 with R software version 3.3.21, employing packages such as stats, made4 [14] and vegan (https://cran.r-project.org/web/packages/vegan/vegan.pdf, accessed on 14 March 2024). The significance of data separation in the PCoA plot was assessed through a permutation test with pseudo-F ratio using the adonis function in the vegan package. Kruskal–Wallis test, along with Wilcoxon tests (paired or unpaired as required), was employed for assessing alpha diversity and relative abundances of taxa. To account for multiple comparisons, *p* values were adjusted using the false discovery rate (FDR) method. *p* values ≤ 0.05 (two-sided *p* value) were considered statistically significant.

## 3. Results

### 3.1. Subjects

Participants in both the intervention and placebo groups ranged in age from 30 to 65 years, with a mean age of 53.5 ± 8.8 and 51.0 ± 10.9 years, respectively. Out of the 204 volunteers, 53 met all inclusion criteria and were randomised into either the intervention or placebo group (Figure 1A,B). Five of the twenty-five participants in the intervention group and three of the twenty-eight participants in the placebo group withdrew from the study for personal reasons. A total of 45 participants completed the study (85% retention). The baseline characteristics including age, sex, anthropometry, body composition, and BP for the enrolled participants are shown in Table 1.

### 3.2. Outcomes

#### 3.2.1. Blood Lipids and Immune/Inflammatory Markers

There were no significant reductions in any of the biochemical parameters, either within or between groups. Although there was a slight reduction in TAG levels in the intervention group from 1.49 ± 0.83 to 1.36 ± 0.69 mmol/L, this change was not statistically significant (*p* = 0.55). Notably, TC and HDL-C levels significantly increased in the placebo group (*p* = 0.005 and *p* = 0.003, respectively). 

The lipid analysis results of both groups are presented in Appendix A, and changes in biochemical parameters from baseline to week 12 are shown in Figure 2A—triaglycerol; Figure 2B—high-density lipoprotein cholesterol (HDL-C); Figure 2C—low-density lipoprotein cholesterol (LDL-C); Figure 2D—non-high-density lipoprotein cholesterol (non-HDL-C); Figure 2E—total cholesterol (TC); and Figure 2F—total cholesterol to HDL-C ratio (THR) as box plots.

#### 3.2.2. Glucose Biomarkers

At the end of the 12-week intervention period, the intervention group showed significant reductions in HbA1c and FPG levels, from 6.06 ± 0.36% to 5.68 ± 0.63%, (*p* = 0.02) and from 6.56 ± 0.77 mmol/L to 6.01 ± 0.76 mmol/L (*p* = 0.003), respectively. Group comparison analysis for HbA1c levels demonstrated a significant difference between the intervention and placebo groups at the end of the intervention period (*p* = 0.03), though this was not the case for FPG levels (*p* = 0.06) (Appendix A). HbA1c and FPG analysis results for both groups are shown in Appendix A, and changes in glucose biomarkers from baseline to week 12 are presented in Figure 3A—fasting plasma glucose; Figure 3B—HbA1c%; Figure 3C—aInsulin; and Figure 3D—HOMA-IR.

#### 3.2.3. Intestinal Permeability

Lipopolysaccharide-binding protein (LBP) was evaluated as a blood serum biomarker of metabolic endotoxemia and, thus, as an indirect biomarker of intestinal barrier function. At the end of the 12-week intervention period, the intervention group showed significant reductions in LPB levels, from 14.8 ± 0.3 to 12.4 ± 0.9, (*p* = 0.002) in comparison to the placebo group as shown in Table 2.

#### 3.2.4. Impact of YMETA Supplementation on the Gut Microbiome

The gut microbiota of the enrolled individuals was profiled over time to assess whether the different gut microbiota layouts and trajectories were associated with YMETA supplementation (Figure 4). Alpha diversity significantly increased in the YMETA group at week 12 (*p* = 0.04, Wilcoxon test; Figure 4A). As for beta diversity, the PCoA of inter-sample variation based on Bray–Curtis dissimilarities (Figure 4B) showed significant segregation between the YMETA and placebo groups, regardless of time point (*p* = 0.01; PERMANOVA), suggesting distinctive features of the gut microbiota among the study groups.

Taxonomic analysis indicated notable differences in core gut microbiome composition (i.e., bacterial genera with relative abundance ≥5% in 20% of the individuals) between the YMETA and placebo groups (Figure 4C). At baseline, the major differences were primarily due to decreased relative abundances of the *Christensenellaceae* R-7 group (*p* = 0.0008; Wilcoxon test) and increased proportions of *Bacteroides* (*p* = 0.004) in the YMETA group. These differences persisted through week 12 and week 16. Additionally, YMETA-specific effects on bacterial genera involved in diabetes revealed a consistent increase over time in the *Eubacterium eligens* group at week 16 compared to baseline and week 12 (*p* ≤ 0.04) (Figure 4D). Although not significant, similar trends were observed for *Barnesiella*, with an increase at week 16 compared to week 12 (*p* = 0.07), and for *Anaerostipes* at week 16 compared to baseline (*p* = 0.09) (Figure 4D).

These findings suggest that YMETA supplementation significantly alters gut microbiota diversity and composition, with potential implications for metabolic health.

#### 3.2.5. Anthropometrics and Body Composition Analysis

In the intervention group, body fat percentage and fat mass decreased slightly from 46.4% to 44% (*p* = 0.06) and from 43.2 kg to 41.9 kg (*p* = 0.2), respectively. Fat-free mass increased from 47.9 kg to 49.7 kg (*p* = 0.113). In the placebo group, body fat percentage and fat mass increased slightly from 38.20% to 40.08% (*p* = 0.2) and from 33.77 kg to 36.25 kg (*p* = 0.2), respectively, with minimal change in fat-free mass from 54.19% to 54.10% (*p* = 0.4). Changes in body composition parameters were not statistically significant in either group (*p* ≥ 0.05). Box plots depicting these changes from baseline to week 12 are shown in Figure 5A—body fat percentage; Figure 5B—fat mass; and Figure 5C—fat-free mass.

#### 3.2.6. Blood Pressure

There was a significant decrease in SBP in the placebo group, from 123 mmHg to 116 mmHg (*p* = 0.006). However, no significant changes in systolic or diastolic BP were found between the two groups (*p* ≥ 0.05). Changes in BP parameters from baseline to week 12 are presented as box plots in Figure 6A—systolic blood pressure and Figure 6B—diastolic blood pressure.

#### 3.2.7. Dietary Intake and Analyses

Table 3 provides information on total daily energy intake (TDEI), macronutrients, and alcohol consumption in grams and as a percentage of TDEI based on participants’ habitual dietary intake. Both groups had a total fibre intake below the recommended 30 [15]. The soluble fibre intake, including starch and sugar, was higher than insoluble fibre intake, which includes non-starch polysaccharides. No statistically significant changes were observed in the either study group.

#### 3.2.8. Gastrointestinal (GI) Changes

No statistically significant changes were observed in stool frequency or form in either study group. Similarly, no statistically significant changes were noted in abdominal pain, bloating, or flatulence throughout the study and none of the individual daily values reported GI symptoms above 1 (present but well tolerated) (Table 4).

## 4. Discussion

This clinical trial investigated the effects of 12 weeks of carbohydrate-based DF supplementation with a GOS-, KGM-, and β-glucan-containing exopolysaccharide (EPS) mixture derived from the probiotic bacterium *Bifidobacterium breve* 091109 on gut health, HbA1c, and FPG in prediabetic subjects aged 18–60 years. The study also examined secondary outcomes including anthropometry, body composition, BP, plasma lipids (TC, HDL-C, LDL-C, TAG, non-HDL-C, and THR), total protein, and dietary intake. After the 12-week intervention, both HbA1c and FPG levels decreased in the intervention group, falling below the ADA classification threshold for prediabetes, whereas the placebo group did not show significant changes. Group comparison analysis showed a significant difference in HbA1c between the intervention and placebo groups at the end of the intervention period, but not in FPG. These findings support previous research indicating that DF can improve glucose metabolism. Mao et al. [16] found that DF supplementation significantly reduced HbA1c by 0.66%, FBG by 0.80 mmol/L, and fasting insulin by 1.68 µIU/mL, with no improvement in BMI, particularly with higher fibre intake (≥10 g/d) and longer intervention periods (8 weeks or more). Soluble fibre has been shown to be more effective in controlling these parameters than a diet containing both soluble and insoluble fibre.

Previous investigations revealed that cardiometabolic risk is higher in prediabetes. Hypertension, dyslipidaemia, and renal risk are highly prevalent in prediabetic adults [17]. According to a meta-analysis, the risk of cardiovascular disease and death is higher in prediabetic people with an FPG concentration as low as 5.6 mmol/L or an HbA1c of 39 mmol/mol (5.7%) compared with those with normal glycaemia [18]. Another study reported that all diagnostic values for prediabetes from different organisations were associated with cardiovascular disease but to different degrees depending on the diagnostic criteria used [19]. This intervention study did not significantly affect plasma or BP within or between the two groups. However, a slight reduction in TAG levels was observed in the intervention group (*p* ≥ 0.05). Previous studies have shown mixed results regarding DF’s effects on cardiometabolic risk factors, with some reporting significant improvements in lipid profiles.

In this study, the clinical dietary intervention combined two types of soluble DF: non-starch polysaccharides (NSP), glucomannan, and exopolysaccharides (EPSs) from *B. breve* 091109 containing β-glucans, and a resistant oligosaccharide, GOS. DFs display prebiotic activity, meaning they do not provide nutrients but enhance beneficial metabolism and promote the growth of probiotic microbes [9] which produce SCFAs, especially acetate, propionate, and butyrate, that affect systemic metabolism and insulin sensitivity [20]. In addition, SCFA production from the fermentation of DF increases intestinal production of GLP-1 and PYY, which improves insulin secretion and, therefore, HbA1c levels [16].

GOS, the most dominant (60%) DF in the formulation, is a non-digestible, soluble fibre with prebiotic properties, promoting the growth of beneficial bacteria such as bifidobacteria and lactobacilli, which produce SCFAs that improve insulin sensitivity and glucose metabolism. GOS also has a low glycaemic index, low calories (1–2 kcal/g), low pH values, and resistance to high temperatures [21]. GOS is divided into α-GOS and β-GOS due to the different galactosidic bonds attached. α-GOSs naturally occur in various plant sources [22]. β-GOSs are derived from the hydrolysis of lactose in an enzymatic manner, catalysed by bacterial beta-galactosidases (lactase). Low concentrations of β-GOSs are also found naturally in milk from animal sources [21,23]. The maximum authorised use of GOS as a food supplement is up to 16.2 g per day [24]. Various studies have evaluated the effects of GOSs in infant nutrition, intestinal health, and dermatology [9,25,26], but studies showing the metabolic effects of GOSs are very limited in the literature. Canfora et al. [27] investigated the effect of supplementing 15 g of GOSs daily for 12 weeks on 44 overweight or obese prediabetic participants. The results showed increased abundance of faecal *Bifidobacterium* species, suggesting that GOS has bifidogenic effects, but it had no significant effect on overall microbial richness or diversity. Also, no significant differences were observed in insulin sensitivity, FPG, TAG, free fatty acids, body composition, BMI, or body weight. In a prior study, Vulevic et al. [28] investigated the effect of daily supplementation of 5.5 g of GOS for 12 weeks on 45 overweight adults. Despite lower GOS intake, GOS supplementation has been shown to have significant effects on insulin, TC, TAG, and TC/HDL-C ratio. Possible reasons for differences between these previous studies and our study may include differences in supplementation doses and composition, as well as demographic and metabolic profiles. Canfora et al. [27] included healthy and overweight or obese elderly participants with prediabetes (mean age 59.2 ± 7.2 years), while Vulevic et al. [28] included both metabolically healthy and overweight younger participants (women: mean age 46.4 ± 11.8; men: mean age 42.8 ± 12.1).

KGM, the second most dominant DF in the formulation, is a water-soluble, highly viscous fermentable fibre derived from the *Amorphophallus konjac* plant. KGM is a high-molecular-weight polysaccharide composed of β-(1 → 4)-linked D-mannose and D-glucose in a 1.6:1.0 ratio. Like GOSs, KGM promotes the growth of the colonic probiotic bacteria bifidobacteria and lactobacilli. The production and absorption of SCFAs from glucomannan are thought to contribute to its metabolic effects, including blood glucose regulation [29]. In addition, the gel-like viscous nature of KGM slows the flow of food in the GI tract, thereby reducing glucose absorption, which may be associated with reduced postprandial blood sugar and increased satiety [30]. The total DF per 100 g of KGM (≥95% based on dry weight) is much higher than that of other DF sources [29,30,31]. The latest scientific review of the health claims regarding KGM and its effect on maintaining normal blood glucose concentrations concluded that a cause-and-effect relationship had not been established. The same conclusion was made for its effect on reducing post-prandial blood glucose response, TAG levels, and maintaining normal gut function [20].

EPSs, present at lower levels (8%) in the formulation, are referred to as microbe-associated molecular patterns (MAMPs). During their growth, bifidobacteria release MAMPs, including capsular polysaccharides (CPSs) and EPSs, which play an important role in maintaining host immune health by modulating intestinal epithelial and immune cells [32]. In both in vitro and in vivo models, EPS-producing *B. breve* UCC2003 has been demonstrated to significantly decrease the production of pro-inflammatory cytokines [33]. The evidence indicates that bifidobacterial molecules prevent colonic tissue damage and inflammation by regulating T-cell responses and strengthening immune tolerance to the existing gut microbiome [34]. In our previous publication [35], we reported the effects of the same EPS postbiotic mixture derived from *Bifidobacterium breve* BB091109, consisting mainly of β-glucan, on plasma markers of systemic inflammation and sex hormones in healthy women over 40 years of age who were not receiving hormone therapy. This study found that daily oral supplementation with a *B. breve* postbiotic extract (500 mg daily) induced protective changes in inflammatory markers, as evidenced by a significant reduction in systemic inflammation markers (CRP, IL-6, TNF-α) after four weeks of supplementation. These changes were maintained throughout the 12-week treatment period and the 4-week follow-up period. This reduction in systemic inflammatory markers was accompanied by a significant decrease in cortisol levels, a hormone associated with increased energy expenditure and immune system activation [36].

β-glucans obtained from algae and bacteria have a linear structure, and glucose monomers are linked via β-(1 → 3) glycosidic bonds, while β-glucans extracted from grains, mushrooms, and yeast have a branched structure, linked via β-(1 → 3) and β-(1 → 6) [37]. The mechanism of the anti-diabetic effects of β-glucans is that a gelatinous layer formed in the intestine reduces the absorption of glucose by enterocytes, thereby reducing glycemia and insulinemia. In addition, glucose absorption reduces with an increase in the thickness of the gelatinous layer [5,38]. Additionally, SCFAs produced from the anaerobic bacterial fermentation of β-glucans in the intestine also contribute to the hypoglycaemic effects. The functionality of β-glucans is related to their physiochemical properties. Most prebiotics used in the food industry are typically composed of oligosaccharides like GOSs, fructo-oligosaccharides, and inulin. However, there is increasing evidence that β-glucans have the potential to be a source of prebiotics and can enhance the probiotic effect which makes it appropriate for developing health-promoting functional foods and supplements [38,39]. Although oat-derived β-glucans have been more extensively researched than β-glucans derived from bacteria, there is evidence that the use of all β-glucans isolated from various sources, regardless of structural differences and molecular weight, has a comparable impact to that of the more commonly used NSP, inulin [40].

Beyond their role in glucose regulation and prebiotic potential, β-glucans and EPSs from probiotic bacteria such as *Bifidobacterium* spp. and *Lactobacillus* spp. function as signalling molecules for epithelial cells. They modulate barrier function through interaction with pattern recognition receptors (PRRs) [41,42]. This signalling influences the integrity and permeability of the intestinal barrier by affecting the expression and organisation of tight junction proteins in epithelial cells, which are crucial for maintaining gut barrier integrity [43]. Additionally, it reduces the production of pro-inflammatory cytokines and increases anti-inflammatory cytokines, helping to control inflammation [44]. The interaction of probiotic EPSs, including β-glucans, with epithelial cell receptors enhances barrier protection, reduces the translocation of harmful pathogens and toxins, and promotes a healthier intestinal environment.

As for the gut microbiome, the significant increase in *Eubacterium eligens* observed over time in the YMETA group, but not in the placebo group, underscores the DF supplement’s potential to beneficially modulate the gut microbiota and confer multiple health benefits. *Eubacterium eligens* has been shown to decrease the risk of type 1 diabetes [45,46,47]. This bacterial species produces SCFAs while degrading pectin and promotes anti-inflammatory effects by enhancing IL-10 production in epithelial cells [40]. Additionally, *Eubacterium eligens* is associated with a beneficial microbial signature for cardiometabolic health, showing consistent benefits across various diets, obesity indicators, and cardiometabolic risks [43]. Its presence also correlates with lower frailty markers, improved cognitive abilities, and increased production of short/branched-chain fatty acids [48,49]. These findings suggest that the YMETA supplement could be a promising non-pharmacological intervention for enhancing gut health and reducing metabolic disorder risks in prediabetic individuals.

The significant increase in *Barnesiella* observed over time in the YMETA group, compared to the placebo group, further suggests the beneficial modulation of the gut microbiota with potential implications for metabolic health. *Barnesiella intestinihominis* has been linked to improved glucose tolerance, possibly through mechanisms involving gut-specific glucose transporter type 1 (GLUT1) inactivation [50]. This bacterial genus is positively correlated with IL-1β levels [51], indicating a potential role in modulating the host’s immune response. The interaction of *Barnesiella* with IL-1 signalling might enhance IL-17 activity, which has further implications for immune regulation [52]. Moreover, *Barnesiella’s* presence is negatively correlated with cerebral small vessel disease (cSVD), suggesting potential neuroprotective effects [53]. These findings underscore the multifaceted benefits of *Barnesiella* enrichment in the gut microbiota, highlighting its relevance in improving glucose metabolism and contributing to broader health benefits, such as immune and brain health, through YMETA supplementation.

Similarly, the significant increase in *Anaerostipes* in the YMETA group over time, compared to the placebo group, highlights the DF supplement’s potential health benefits. *Anaerostipes* is known to be reduced in patients with T2D and other metabolic disorders such as inflammatory bowel disease (IBD) and irritable bowel syndrome (IBS) [54,55]. The presence of this genus is inversely associated with insulin resistance, as higher levels of *Anaerostipes* correlate with lower HOMA-IR scores [52]. Additionally, *Anaerostipes* is a major butyrate producer in the human gut [56,57,58], contributing to the production of SCFAs that are crucial for maintaining gut health and metabolic functions. The increase in *Anaerostipes* in the YMETA group suggests that the DF supplement may enhance butyrate production, thereby improving insulin sensitivity and reducing metabolic disorder risks. These findings underscore the therapeutic potential of the YMETA supplement in modulating gut microbiota to confer significant metabolic health benefits.

Studies examining the effect of prebiotics and bioactive compounds on metabolic markers in prediabetes are scarce. A particular strength of this study is that, to our knowledge, there have been no prior investigations in the literature that consider the metabolic effects of combining GOSs, KGM, and β-glucans. These fibres have previously been investigated individually or in combination with other prebiotics or probiotics. The use of HbA1c as a diagnostic method is another strength of the current study; since this is a non-fasting test, it is less influenced by factors such as stress or infection and it has high specificity for the diagnosis of prolonged hyperglycaemia [3]. The double-blind, randomised, placebo-controlled, and parallel-grouped study design and the adequate intervention period further strengthen the findings.

However, some limitations should be noted. Diet and nutritional status are among the most important, modifiable determinants of human health. The nutritional value of food is influenced in part by a person’s gut microbial community (microbiota) and its component genes (microbiome). Unravelling the interrelationships between diet, the structure and operations of the gut microbiota, and nutrient and energy harvest is confounded by variations in human environmental exposures, microbial ecology and genotype. For example, not all participants completed the washout period and data were missing to determine whether the effects of supplementation wore off after the end of the intervention period. Furthermore, food diaries were not requested at the end of the intervention period to monitor potential dietary changes and provide evidence of whether the observed effects were due to the supplement or dietary changes.

The current study formulation also may not provide a clear dose–response pattern and suggest that the supplementation dosage might be insufficient.

## 5. Conclusions

In conclusion, in this clinical trial, after 12 weeks of DF supplementation containing GOSs, KGM and β-glucans, a statistically significant reduction was observed in HbA1c and FPG levels in the intervention group, while no significant change was observed in the placebo group. No significant reductions were observed in other blood parameters (e.g., plasma lipids and total protein) or anthropometric measurements (e.g., body composition and BP) within or between groups. The analysis of habitual dietary intake revealed that the total daily fibre intake of both groups was lower than the recommended daily intake and that the intake of soluble fibre was higher compared to insoluble fibre in both groups. The DF supplementation significantly increased the abundance of the beneficial gut bacteria *Eubacterium eligens*, *Barnesiella*, and *Anaerostipes*, suggesting its potential to enhance gut health, improve glucose metabolism, and confer broader metabolic, immune, and neuroprotective benefits in prediabetic individuals.

## Figures and Tables

**Figure 1 nutrients-16-02205-f001:**
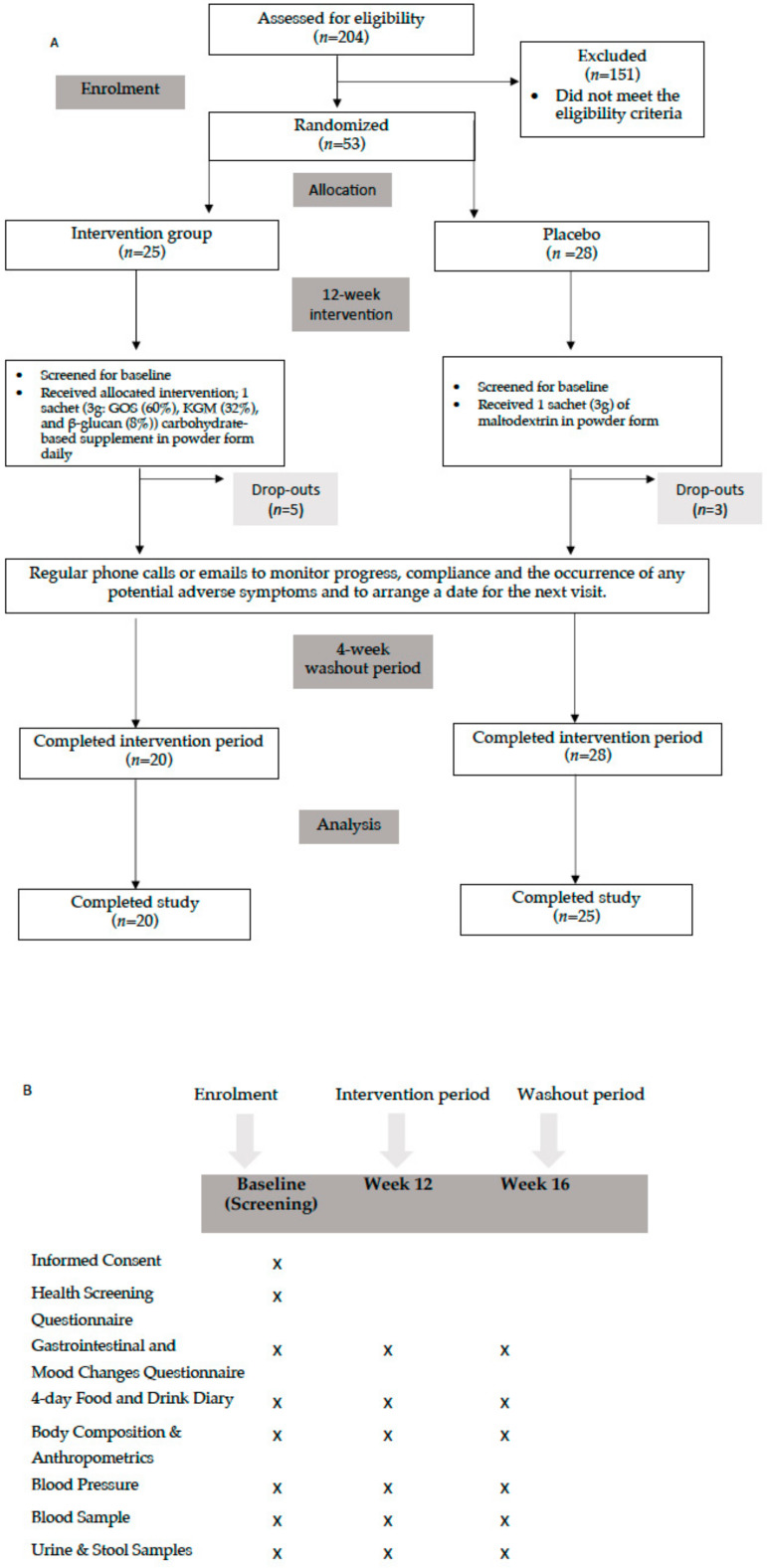
(**A**) Flow Diagram of the study; (**B**) outcomes measured at different time points of the study.

**Figure 2 nutrients-16-02205-f002:**
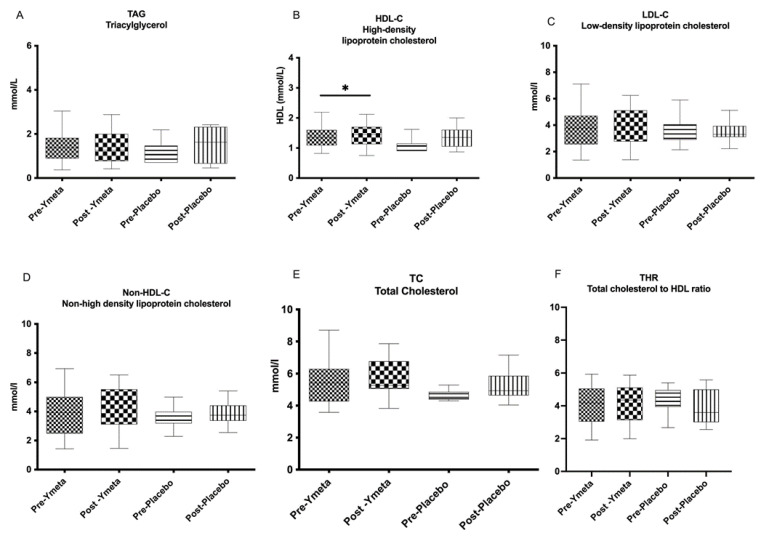
Changes in biochemical parameters from baseline to week 12. (**A**) Triaglycerol; (**B**) high-density lipoprotein cholesterol (HDL-C); (**C**) low-density lipoprotein cholesterol (LDL-C); (**D**) non-high-density lipoprotein cholesterol (non-HDL-C; (**E**) total cholesterol (TC); (**F**) total cholesterol to HDL-C ratio (THR). Data are presented as box plots with maximum value, 75th percentile (upper line of box), median (middle line of box), 25th percentile (lower line of box), and minimum. *p* value less than 0.05 (* two-sided *p* value) was considered statistically significant.

**Figure 3 nutrients-16-02205-f003:**
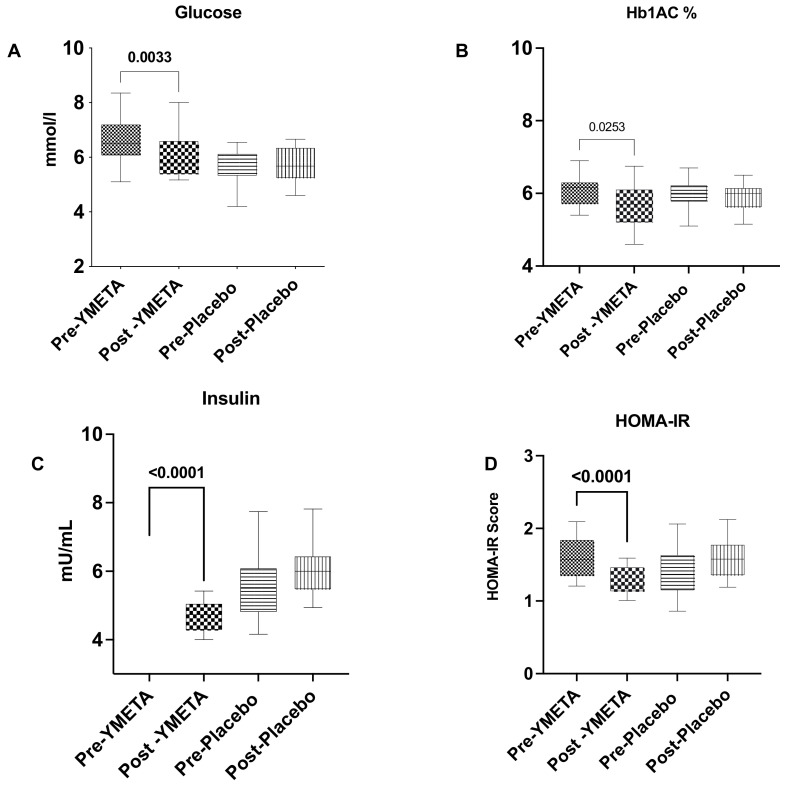
Changes in glucose biomarkers from baseline to week 12. (**A**) Fasting plasma glucose; (**B**) HbA1c%; (**C**) insulin; (**D**) HOMA-IR. Data are presented as box plots with maximum value, 75th percentile (upper line of box), median (middle line of box), 25th percentile (lower line of box), and minimum. *p* value less than 0.05 (two-sided *p* value) was considered statistically significant.

**Figure 4 nutrients-16-02205-f004:**
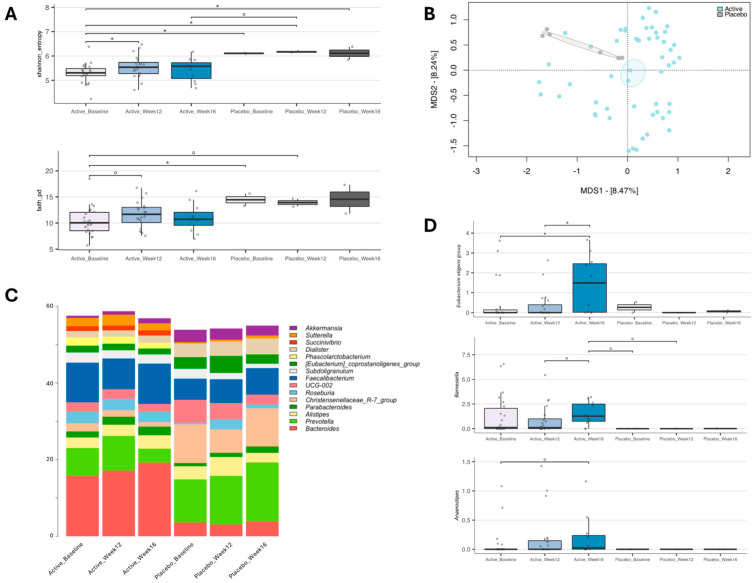
Gut microbiome dynamics of enrolled individuals supplemented with YMETA vs. the placebo. (**A**) Box plots showing the distribution of alpha diversity, computed with the Shannon index (shannon_entropy; top) and Faith’s Phylogenetic Diversity (faith_pd; bottom), for faecal samples collected at baseline, week 12, and week 16. Wilcoxon test, * *p* value ≤ 0.05; ° *p* value ≤ 0.1. (**B**) A Principal Coordinates Analysis (PCoA) based on the Bray–Curtis dissimilarities between microbiota profiles. A significant separation was found between the YMETA and placebo groups (PERMANOVA, *p* = 0.01). (**C**) Genus-level relative abundance profiles of the gut microbiota of enrolled individuals supplemented with YMETA or the placebo at baseline and after 12 and 16 weeks. Bar plots show the average relative abundance of the core microbiome at the genus level (i.e., 5% in at least 20% of the individuals) for each group. (**D**) Box plots showing the relative abundance distribution of bacterial genera differentially represented in the gut microbiota of the study groups. Only bacterial genera relevant to diabetes that showed significant increases or decreases in the YMETA groups are displayed. Wilcoxon test, * *p* ≤ 0.05; ° *p* ≤ 0.01.

**Figure 5 nutrients-16-02205-f005:**
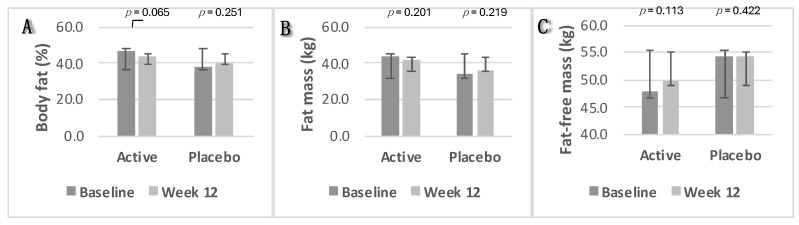
Changes in body composition parameters. (**A**) Body fat percentage; (**B**) fat mass; (**C**) fat-free mass. Data were analysed using Paired Samples *t*-Test. *p* value less than 0.05 (two-sided *p* value) was considered statistically significant. Data are presented as histograms with mean values and with error bars indicating standard deviations.

**Figure 6 nutrients-16-02205-f006:**
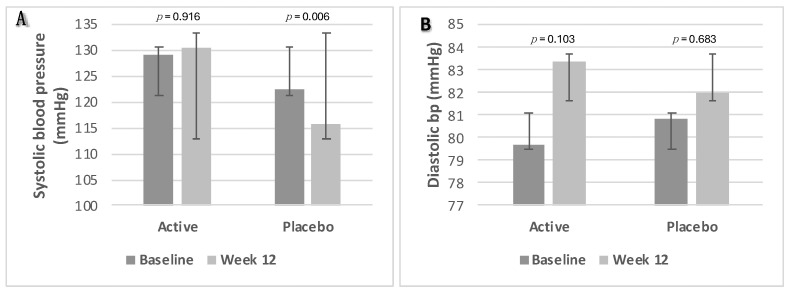
Changes in blood pressure parameters. (**A**) Systolic blood pressure; (**B**) diastolic blood pressure. Data were analysed using Paired Samples *t*-Test. *p* value less than 0.05 (two-sided *p* value) was considered statistically significant. Data are presented as histograms with mean values and with error bars indicating standard deviations.

**Table 1 nutrients-16-02205-t001:** Participant characteristics at the baseline visit.

Descriptives	Intervention (*n* = 25)	Placebo (*n* = 25)	*p*-Value
*Age (years)*	53.5 ± 8.8	51.0 ± 10.9	0.369
Sex (M/F), *n/n*	7/18	16/9	
*Anthropometric Measurements*			
Height (cm)	167.6 ± 8.8	172.1 ± 10.5	0.095
Weight (kg)	91.9 ± 20.8	87.5 ± 17.5	0.414
Body Mass Index (kg/m^2^)	32.4 ± 6.3	29.0 ± 4.0	0.019
Waist Circumference (cm)	100.2 ± 13.7	106.5 ± 30.2	0.342
Hip Circumference (cm)	114.1 ± 13.4	101.7 ± 21.4	0.015
Waist/Hip Ratio (waist/hip circumference)	0.88 ± 0.07	1.24 ± 1.03	0.085
*Body Composition*			
Body Fat (%)	46.4 ± 9.4	38.3 ± 7.1	<0.001
Fat Mass (kg)	43.8 ± 17.8	33.7 ± 9.9	0.013
Fat Free Mass (kg)	47.8 ± 8.4	53.9 ± 12.2	0.043
*Blood Pressure*			
Systolic Blood Pressure (mmHg)	130.5 ± 12.7	122.7 ± 7.9	0.009
Diastolic Blood Pressure (mmHg)	80.2 ± 8.4	80.8 ± 10.2	0.814

The data are expressed as Mean ± Standard Deviation (SD). The data were analysed using the Independent Samples *t*-Test. A *p* value less than 0.05 (two-sided *p* value) was considered statistically significant.

**Table 2 nutrients-16-02205-t002:** Inflammatory parameters at baseline and post-intervention (week 12) in YMETA-treated and control (placebo comparator) groups.

Variable LBP, μg/mL	Baseline	Post-Intervention	Treatment	Time	Interaction
Intervention	14.8 ± 0.3	12.4 ± 0.9	0.732	0.002 *	0.207
Control	18.7 ± 0.6	15.4 ± 1.1			0.6

Values are mean ± SEM. *p* values were calculated using repeated measures ANOVA. * *p* ≤ 0.05 was considered statistically significant. Abbreviation: LBP, lipopolysaccharide-binding protein.

**Table 3 nutrients-16-02205-t003:** Habitual dietary intake of the participants.

Descriptives	Intervention (*n* = 20)	Placebo (*n* = 25)
Total Daily Energy Intake (kcals)	1769.9 ± 525.8	2566.7 ± 964.3
Protein (g) (% energy)	78.3 ± 29.9 (18.1 ± 5.9)	93.5 ± 33.2 (15.0 ± 1.5)
Carbohydrate (g) (% energy)	194.4 ± 54.2 (43.5 ± 8.5)	303.4 ± 103.3 (43.1 ± 2.7)
Dietary Fibre (g)	20.5 ± 7.4	27.1 ± 9.0
Soluble Fibre (g)		
Starch (g)	109.4 ± 54.2	207.3 ± 77.9
Sugars (g)	81.5 ± 27.5	94.8 ± 37.3
Insoluble Fibre (g)		
Non-Starch Polysaccharides (NSPs) (g)	15.7 ± 3.9	19.5 ± 6.8
Fat (g) (% energy)	72.8 ± 25.8 (35.6 ± 5.4)	128.9 ± 63.7 (39.7 ± 19.6)
Alcohol (% energy)	7.2 ± 10.1 (2.7 ± 4.0)	8.2 ± 9.3 (2.8 ± 4.9)

The data are expressed as mean ± standard deviation. The amount of macro and micronutrients is given as grams (g) and percentages (% energy) of the total daily energy intake.

**Table 4 nutrients-16-02205-t004:** Gastrointestinal symptom rating scale (GSRS) questionnaire data for each symptom at baseline and post-intervention in YMETA-treated and control (placebo comparator) groups.

Variable	Baseline	Post-Intervention	Treatment	Time	Interaction
Abdominal discomfort					
Intervention	1.18 ± 0.36	1.09 ± 0.34	0.938	0.0091	0.439
Control	1.67 ± 0.53	1.78 ± 0.56			
Heartburn					
Intervention	1.27 ± 0.38	1.00 ± 0.31	0.067	0.583	0.884
Control	1.33 ± 0.42	1.11 ± 0.35			
Acid reflux					
Intervention	1.09 ± 0.33	1.18 ± 0.374	0.810	0.224	0.380
Control	1.00 ± 0.31	1.01 ± 0.34			
Hunger pains					
Intervention	1.2 ± 0.3	1.1 ± 0.2	0.817	0.068	0.842
Control	1.2 ± 0.2	1.3 ± 0.2			
Nausea					
Intervention	1.00 ± 0.30	1.55 ± 0.49	0.988	0.180	0.475
Control	2.33 ± 0.73	2.11 ± 0.66			
Rumbling					
Intervention	1.64 ± 0.49	1.55 ± 0.49	0.558	0.057	0.805
Control	2.33 ± 0.74	2.11 ± 0.67			
Bloating					
Intervention	2.45 ± 0.74	2.00 ± 0.63	0.152	0.693	0.614
Control	2.11 ± 0.66	1.90 ± 0.59			
Burping					
Intervention	1.18 ± 0.36	1.45 ± 0.46	0.152	0.614	0.263
Control	1.78 ± 0.56	1.33 ± 0.42			
Flatus					
Intervention	1.27 ± 0.38	1.55 ± 0.49	0.815	0.334	0.988
Control	2.33 ± 0.73	1.89 ± 0.59			
Constipation					
Intervention	1.28 ± 0.42	1.27 ± 0.40	0.161	0.161	0.879
Control	1.56 ± 0.49	1.11 ± 0.35			
Diarrheal					
Intervention	1.18 ± 0.36	1.45 ± 0.46	0.747	0.187	0.555
Control	1.78 ± 0.56	1.33 ± 0.42			
Loose stools					
Intervention	1.36 ± 0.41	1.82 ± 0.57	0.569	0.569	0.336
Control	1.67 ± 0.52	1.72 ± 0.64			
Hard stools					
Intervention	1.45 ± 0.44	1.09 ± 0.34	0.448	0.162	0.129
Control	1.00 ± 0.31	1.13 ± 0.35			
Bowel movement					
Intervention	1.73 ± 0.52	2.00 ± 0.63	0.387	0.929	0.750
Control	2.00 ± 0.63	2.38 ± 0.75			
Not complete bowel emptiness sensation					
Intervention	2.09 ± 0.63	1.91 ± 0.60	0.103	0.481	0.518
Control	1.78 ± 0.56	1.38 ± 0.43			

Values are expressed as Mean ± SEM. YMETA intervention. *p* values calculated using repeated measures ANOVA.

## Data Availability

The data presented in this study are available upon request from the corresponding author (due to legal and ethical reasons).

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
