# Peer review of "Impact of Combined Prebiotic Galacto-Oligosaccharides and Bifidobacterium breve-Derived Postbiotic on Gut Microbiota and HbA1c in Prediabetic Adults: A Double-Blind, Randomized, Placebo-Controlled Study"

_nutrients, 2024, doi:10.3390/nu16142205_

Round 1

Reviewer 1 Report

Comments and Suggestions for Authors

The paper is well written and interesting. It represents a lot of work and has generally been well conducted.

However the authors could make a few changes:
Line 69 "-- ameliorate glucose homeostasis -- should be replaced by " --- ameliorate glucose intolerance --".
Line 70: A little more introduction on specifically why the materials investigated were selected, and some clear hypotheses regarding the possible outcomes.
Line 105: How were the volunteers recruited?
Line 118: maltodextrin was given as a placebo - please cklarify whether this was digestible ( therefore glycaemic) or non-digestible (prebiotic) maltodextrin.

Was there a power analysis done to guide subject selection?
Because of the rapid effect of diet on the microbiome  an analysis of food intake closer to the 12 week end point would have been valuable. The authors should perhaps discuss this as a limitation of their study.
The possible effect of the regular monitoring on subjective measurements should perhaps be mentioned.

The axis labels on the graphs are too small to be readable in many cases.

The discussion is interesting and to the point

Although there were no significant effects on many of the outcomes it is nonetheless important to publish non-significant findings.

Comments on the Quality of English Language

--

Author Response

REVIEWER 1

Comment 1: Line 69 "-- ameliorate glucose homeostasis -- should be replaced by " --- ameliorate glucose intolerance --".

Response1: We thank the reviewer for highlighting this comment, and we have now changed it. ‘These supplements promote the growth of beneficial bacteria, but may also mitigate inflammation, enhance barrier function, and ameliorate glucose intolerance [8,9].’

Comment 2: Line 70: A little more introduction on specifically why the materials investigated were selected, and some clear hypotheses regarding the possible outcomes.

Response 2: We thank the reviewer for highlighting this comment, and we have now changed it – Lines 76-83 – “Firstly, we conducted a comprehensive assessment of gut health, including gut microbiota composition and intestinal permeability, as well as metabolic function, specifically glycated haemoglobin (HbA1c) and fasting plasma glucose (FPG) levels. Then, we evaluated alterations in anthropometry, body composition, blood pressure (BP), and plasma lipids [total cholesterol (TC), HDL-C, LDL-C, triacylglycerol (TAG), non-HDL-C and TC to HDL-C ratio (THR)]. Additionally, we examined total protein and diet to elucidate the impact of DF supplementation on prediabetes management and to inform evidence-based preventative strategies.’

Comment 3: Line 105: How were the volunteers recruited?

Response 3: Please read the following change – Lines 114-115 – ‘Two hundred four volunteers were recruited from the local community and social media platforms such as Facebook or Instagram, which were widely used among adults’.

Comment 4: Line 118: maltodextrin was given as a placebo - please clarify whether this was digestible (therefore glycaemic) or non-digestible (prebiotic) maltodextrin.

Response 4: We have added the following explanation – Lines 128-130 – ‘Cellulose microcrystalline (Alfa Aesar, UK), a common excipient in the pharmaceutical industry known for its lack of impact in the colonic environment, was chosen as the placebo.’

Comment 5: Was there a power analysis done to guide subject selection?

Response 5: We thank the reviewer for this comment, and we would like to confirm that the study was performed as summarised in Figure 1. A total of 204 volunteers were contacted for participation in the study and received a detailed explanation of the study design and protocol. All prospective volunteers (n=53) were screened by the same investigator and among these, fulfilled the inclusion criteria and agreed to take part in the study. Out of 53 participants, only 8 participants dropped out due personal matters. Moreover, we also would like to confirm that the statistical power of the study sample was calculated considering a total of 53 participants. 

Comment 6: Because of the rapid effect of diet on the microbiome an analysis of food intake closer to the 12 week end point would have been valuable. The authors should perhaps discuss this as a limitation of their study.

Response 6: We thank the reviewer for highlighting this comment, and we have now changed it with more emphasis on the impact of the diet to influence the relationship between gut microbiota and individual health outcomes. Lines 314-336 – 3.2.4. Impact of YMETA Supplementation on the Gut Microbiome

Gut microbiota profiling revealed significant changes in diversity and composition. The gut microbiota of the enrolled individuals was profiled over time to assess whether the different gut microbiota layouts and trajectories were associated with YMETA supplementation (Figure 4). Alpha diversity significantly increased in the YMETA group at week 12 (P= 0.04, Wilcoxon test; Figure 4A). As for beta diversity, the PCoA of inter-sample variation based on Bray-Curtis dissimilarities (Figure 5B) showed significant segregation between YMETA and placebo, regardless of time point (P = 0.01; PERMANOVA), suggesting distinctive features of the gut microbiota among the study groups.

Taxonomic analysis indicated notable differences in core gut microbiome composition (i.e., bacterial genera with relative abundance ≥5% in 20% of individuals) between the YMETA and placebo groups (Figure 4C). At baseline, the major differences were primarily due to decreased relative abundances of Christensenellaceae R-7 group (p = 0.0008; Wilcoxon test) and increased proportions of Bacteroides (P = 0.004) in the YMETA group. These differences persisted through week 12 and week 16. Additionally, YMETA-specific effects on bacterial genera involved in diabetes revealed a consistent increase over time in Eubacterium eligens group at week 16 compared to baseline and week 12 (P ≤ 0.04). Although not significant, similar trends were observed for Barnesiella, with an increase at week 16 compared to week 12 (P= 0.07), and for Anaerostipes at week 16 compared to baseline (P = 0.09).

These findings suggest that YMETA supplementation significantly alters gut microbiota diversity and composition, with potential implications for metabolic health.’

Lines 516-564 – ‘Beyond their role in glucose regulation and prebiotic potential, β-glucans and EPS from probiotic bacteria such as Bifidobacterium spp. and Lactobacillus spp. function as signaling molecules for epithelial cells. They modulate barrier function through interaction with pattern recognition receptors (PRRs) [41, 42]. This signaling influences the integrity and permeability of the intestinal barrier by affecting the expression and organisation of tight junction proteins in epithelial cells, which are crucial for maintaining gut barrier integrity [43]. Additionally, it reduces the production of pro-inflammatory cytokines and increases anti-inflammatory cytokines, helping to control inflammation [44]. The interaction of probiotic EPS, including β-glucans, with epithelial cell receptors enhances barrier protection, reduces the translocation of harmful pathogens and toxins, and promotes a healthier intestinal environment.

As for the gut microbiome, the significant increase in Eubacterium eligens group observed over time in the YMETA group, but not in the placebo group, underscores the DF supplement's potential to beneficially modulate the gut microbiota and confer multiple health benefits. Eubacterium eligens has been shown to decrease the risk of type 1 diabetes [45, 46, 47]. This bacterial species produces SCFAs while degrading pectin and promotes anti-inflammatory effects by enhancing IL-10 production in epithelial cells [40]. Additionally, Eubacterium eligens is associated with a beneficial microbial signature for cardiometabolic health, showing consistent benefits across various diets, obesity indicators, and cardiometabolic risks [43]. Its presence also correlates with lower frailty markers, improved cognitive abilities, and increased production of short/branched chain fatty acids [48, 49]. These findings suggest that the YMETA supplement could be a promising non-pharmacological intervention for enhancing gut health and reducing metabolic disorder risks in prediabetic individuals.

The significant increase in Barnesiella observed over time in the YMETA group, compared to the placebo group, further suggests beneficial modulation of the gut microbiota with potential implications for metabolic health. Barnesiella intestinihominis has been linked to improved glucose tolerance, possibly through mechanisms involving gut-specific glucose transporter type 1 (GLUT1) inactivation [50]. This bacterial genus is positively correlated with IL-1β levels [51], indicating a potential role in modulating the host's immune response. The interaction of Barnesiella with IL-1 signaling might enhance IL-17 activity, which has further implications for immune regulation [52]. Moreover, Barnesiella's presence is negatively correlated with cerebral small vessel disease (cSVD), suggesting potential neuroprotective effects [53]. These findings underscore the multifaceted benefits of Barnesiella enrichment in the gut microbiota, highlighting its relevance in improving glucose metabolism and contributing to broader health benefits, such as immune and brain health, through YMETA supplementation.

Similarly, the significant increase in Anaerostipes in the YMETA group over time, compared to the placebo group, highlights the DF supplement's potential health benefits. Anaerostipes is known to be reduced in patients with T2D and other metabolic disorders such as inflammatory bowel disease (IBD) and irritable bowel syndrome (IBS) [54,55]. The presence of this genus is inversely associated with insulin resistance, as higher levels of Anaerostipes correlate with lower HOMA-IR scores [52]. Additionally, Anaerostipes is a major butyrate producer in the human gut [56-58], contributing to the production of SCFAs that are crucial for maintaining gut health and metabolic functions. The increase in Anaerostipes in the YMETA group suggests that the DF supplement may enhance butyrate production, thereby improving insulin sensitivity and reducing metabolic disorder risks. These findings underscore the therapeutic potential of the YMETA supplement in modulating gut microbiota to confer significant metabolic health benefits.’

Comment 7: The possible effect of the regular monitoring on subjective measurements should perhaps be mentioned.

Response 7: We thank the reviewer for highlighting this comment and we have now changed it.

Comment 8: The axis labels on the graphs are too small to be readable in many cases.

Response 8: All figures have been now added with a better resolution.

Reviewer 2 Report

Comments and Suggestions for Authors

This manuscript presents the results obtained from research performed to evaluate the therapeutic potential during a 12-week dietary-fibre (DF) supplement, named Y-META, based on galactooligosaccharides (GOS) produced through the β-galactosidase activity of Bifidobacterium breve, exopolysaccharides (EPS) from B. breve, and konjac glucomannan (KGM), in prediabetic patients aged 18-60 years. 

During the clinical trial, after 12 weeks of dietary fibre supplementation, a significant reduction in HbA1c  and FPG levels in the intervention group compared to the placebo group. Molecular analysis performed revealed the increased population of beneficial microorganisms, such as  Eubacterium eligens,  Barnesiella, and Anaerostipes in the human gut microbiota. These last results reveal the potential of dietary fibre used, to modulate the gut microbiota and reduce metabolic disorders.

The article is well written, experiments were properly conducted, and the results obtained are interesting. However, the manuscript cannot published in this form, because more corrections are still needed, as follows:

 1) at row 43, the punctuation sign must be removed;

2) in the manuscript exist abbreviations without explanations ( HbA1c, LPS;  TAG/5;   FPG; ADA; TAG; TC; T2D etc). In scientific articles, a short explanation must be added in the bracket, when an abbreviation was used for the first time in the text.  If these explanations are too long,  then at the end of the manuscript, authors must add a Chapter entitled ''Abbreviations'' in which these abbreviations will be explained.

3) Figure 4 missing; for this reason, Figure  5 becomes Figure  4; Figure 6 becomes Figure 5, and Figure 7 becomes Figure 6. Proper corrections must be done in the text too. 

4. In the manuscript more sentences begin with an abbreviation; these 

sentences must be rewritten.

5) The references must be written according to MDPI rules.

6) The scientific name of all microorganisms   must be written in italics, without brackets (i.e. instead of [Eubacterium] eligens, must be written Eubacterium eligens).

7) Figure 5C (after correction will become Figure 4C) is not visible; authors must provide here a picture with a better resolution.

Author Response

REVIEWER 2

Comment 1: at row 43, the punctuation sign must be removed.

Response 1: Thank you and now it has been corrected.

Comment 2: in the manuscript exist abbreviations without explanations ( HbA1c, LPS;  TAG/5;   FPG; ADA; TAG; TC; T2D etc). In scientific articles, a short explanation must be added in the bracket, when an abbreviation was used for the first time in the text.  If these explanations are too long,  then at the end of the manuscript, authors must add a Chapter entitled ''Abbreviations'' in which these abbreviations will be explained.

Response 2: Apologies for this point and now we have added the full explanation throughout the manuscript.

Comment 3: Figure 4 missing; for this reason, Figure  5 becomes Figure  4; Figure 6 becomes Figure 5, and Figure 7 becomes Figure 6. Proper corrections must be done in the text too. 

Response 3: This has been revised accordingly.

Comment 4: In the manuscript more sentences begin with an abbreviation; these sentences must be rewritten.

Response 4: Apologies for this point as previously mentioned it is now added.

Comment 5: The references must be written according to MDPI rules.

Response 5: All references have been checked and in line with the journal style.

Comment 6: The scientific name of all microorganisms   must be written in italics, without brackets (i.e. instead of [Eubacterium] eligens, must be written Eubacterium eligens).

Response 6: It is now corrected and changed.

Comment 7: Figure 5C (after correction will become Figure 4C) is not visible; authors must provide here a picture with a better resolution.

Response 7: It is now uploaded and changed.

Reviewer 3 Report

Comments and Suggestions for Authors

The present study could be intriguing if it had been performed in patients with Diabetes Mellitus type 2 and received the formula. It isn't easy to evaluate the effect of any treatment in prediabetic adults. The authors of that term mean the IGT and IFT states.

According to the literature, studies that evaluated IFG and IGT states over time found that some individuals will become normal again. Which percentage of participants normalized, and what is the percentage in other studies with normalization of glucose indices?

Moreover, the authors wrote that they used an HbA1c level of 5.7 to 6.4% (based on the ADA criteria) to define the IGT and IFG state. However, the test should be performed in a laboratory using the National Glycohemoglobin Standardization Program (NGSP) method certified and standardized to the DCCT assay. [Diabetes Care 2024;47(Supplement 1): S20–S42]. Did they use that method? Otherwise, the values of HbA1c could be false.

I did not find data about the microvascular complications of DM, such as Neuropathy and retinopathy. Those could be observed in Prediabetes.

Furthermore, how did the author conclude that an 18-year-old individual had type 2 DM?

Data about treatment for hypertension and hyperlidemia are missing. The authors observed changes in total cholesterol and HDL in the placebo group. Was there any change in treatment?

Overall, the data are interesting, but the authors should discuss the abovementioned comments.

Comments on the Quality of English Language

none

Author Response

REVIEWER 3

Comment 1: The present study could be intriguing if it had been performed in patients with Diabetes Mellitus type 2 and received the formula. It isn't easy to evaluate the effect of any treatment in prediabetic adults. The authors of that term mean the IGT and IFT states.

Response 1: That may be true, but performing the study in Type 2 diabetes patients would be a disease treatment and the formulation would not qualify for medicinal use. This is a dietary intervention study, aiming to study whether a gut health focused supplement would be able to impact markers of metabolic function in people at risk of developing type 2 diabetes (i.e pre-diabetes).

Comment 2: According to the literature, studies that evaluated IFG and IGT states over time found that some individuals will become normal again. Which percentage of participants normalized, and what is the percentage in other studies with normalization of glucose indices?

Response 2: A meta-analysis of various studies indicated that the rate of reversion to normal glucose levels can be as high as 30% in certain populations, particularly when baseline glucose levels are closer to normal and less severe (American Diabetes Association: Impaired Glucose Tolerance and Impaired Fasting Glucose). However, factors such as younger age, lower baseline glucose levels, and fewer comorbidities increase the likelihood of reversion to normal glucose levels. Additionally, incidental lifestyle changes, such as improved diet and increased physical activity, can contribute to these outcomes even without formal interventions.

Comment 3: Moreover, the authors wrote that they used an HbA1c level of 5.7 to 6.4% (based on the ADA criteria) to define the IGT and IFG state. However, the test should be performed in a laboratory using the National Glycohemoglobin Standardization Program (NGSP) method certified and standardised to the DCCT assay. [Diabetes Care 2024;47(Supplement 1): S20–S42]. Did they use that method? Otherwise, the values of HbA1c could be false.

Response 3: We thank the Reviewer for this point, and we agree that there has been considerable improvement in the quality of HbA1c testing since the NGSP was initiated over 20 years ago. Virtually all laboratories in developed countries now report all glycated hemoglobin results exclusively as HbA1c. Moreover, overall variability within and among methods and among individual laboratories has been considerably reduced. In 2009, international experts from the ADA, the IDF, and the EASD joined to recommend the use of HbA1c for the diagnosis of diabetes. This recommendation was due in large part to advances in HbA1c assay standardization. The NycoCard™ D-dimer assay (“NycoCard”) (Abbott Laboratories, Copenhagen, Denmark), is based on an immunometric flow through principle, with plasma d-dimer molecules being trapped on a surface membrane with d-dimer specific monoclonal antibodies. The addition of a secondary labelled antibody generates a subsequent color development, with the color intensity being proportional to the d-dimer concentration. The assay has a measuring range of 0.1–20 mg/L, a calibrated human assay range of 0.1–10.0 mg/L and a measuring interval of 0.1 mg/L (NycoCardTM D-dimer, technical support, Abbott Laboratories, Copenhagen, Denmark). Measurements displayed by the machine as < 0.1 mg/L in this study were converted to 0.09 mg/L for statistical purposes. We can confirm that this methodology is validated against the National Glycohemoglobin Standardization Program (NGSP).

Comment 4: I did not find data about the microvascular complications of DM, such as Neuropathy and retinopathy. Those could be observed in Prediabetes. Furthermore, how did the author conclude that an 18-year-old individual had type 2 DM?

Response 4: Retinopathy, nephropathy and neuropathy (microvascular complications) impact hundreds of millions of diabetics and normally target those having long-term or uncontrolled forms of the disease; however, these disorders can also exist at the time of diagnosis or in those yet to be diagnosed. That was not the focus of the study. Diabetes comorbid conditions are part of the exclusion criteria. The population is not diabetics, they are healthy at risk of developing diabetes, but diabetes related complications are not present yet. In addition the study was relatively short, focused on HbA1c time requirements for meaningful change in a 3 months window. We thank the Reviewer for this comment.

Comment 5: Data about treatment for hypertension and hyperlidemia are missing. The authors observed changes in total cholesterol and HDL in the placebo group. Was there any change in treatment?

Response 5: There was a slight reduction in TAG levels in the intervention group from 1.49±.83 to1.36±.69 mmol/L, this change was not statistically significant (P=0.55). Notably, TC and HDL-C levels significantly increased in the placebo group (P=0.005 and P=0.003, respectively). The lipid analysis results of both groups are presented in Table S1, and changes in biochemical parameters from baseline to week 12 are shown in Figure 2 as box plots.

Comment 6: Overall, the data are interesting, but the authors should discuss the abovementioned comments.

Response 6: All these points have been now clarified and addressed.

Round 2

Reviewer 2 Report

Comments and Suggestions for Authors

Generally, the authors have repaired the major discrepancies in their manuscript. However, the current variant still needs more attention, regarding the following aspects:

The Table S2 become Table 2 and the rest of the Tables must be renamed, (Table 2 becomes Table 3; Table 3 becomes Table 4, Table 4 becomes Table 5) 

All pictures must be mentioned in the text; respectively

-Figure 1B must be  mentioned in the text;

-Regarding Figure 2: authors must put on each  figure the  corresponding letters: A, B, C, D E, F. After that, Figure 2A, figure 2B, figure 2C, figure 2D, figure 2E, and Figure 2F will be mentioned in the text;

-Figure 3A, figure 3B, figure 3C and figure 3D must be mentioned in the text;

-At row 305 instead of Figure 5B authors must write Figure 4B; also, Figure 4D must be mentioned in the text. In the figure 4D, at the Oy axis, instead of [Eubacterium]_ eligens group authors must write Eubacterium eligens group

-Figure 5A, figure 5B and figure 5C must appear in the text;

-Figure 6A and Figure 6B must appear in the text.

Author Response

The Table S2 become Table 2 and the rest of the Tables must be renamed, (Table 2 becomes Table 3; Table 3 becomes Table 4, Table 4 becomes Table 5) 

Response This point have been corrected in the text.

All pictures must be mentioned in the text; respectively

-Figure 1B must be  mentioned in the text;

Response this has been corrected in the text

-Regarding Figure 2: authors must put on each  figure the  corresponding letters: A, B, C, D E, F. After that, Figure 2A, figure 2B, figure 2C, figure 2D, figure 2E, and Figure 2F will be mentioned in the text;

Response These  points have been corrected in the text

-Figure 3A, figure 3B, figure 3C and figure 3D must be mentioned in the text;

Response These  points have been corrected in the text

-At row 305 instead of Figure 5B authors must write Figure 4B; also, Figure 4D must be mentioned in the text. In the figure 4D, at the Oy axis, instead of [Eubacterium]_ eligens group authors must write Eubacterium eligens group

Response y axis now it is changed and reads Eubacterium eligens group

-Figure 5A, figure 5B and figure 5C must appear in the text;

Response This point have been corrected in the text

-Figure 6A and Figure 6B must appear in the text.

Response This point have been corrected in the text

Reviewer 3 Report

Comments and Suggestions for Authors

no other comments

Comments on the Quality of English Language

none

Author Response

This reviewer does not have any additional comments